# In Situ SABRE Hyperpolarization with Earth’s Field NMR Detection

**DOI:** 10.3390/molecules24224126

**Published:** 2019-11-14

**Authors:** Fraser Hill-Casey, Aminata Sakho, Ahmed Mohammed, Matheus Rossetto, Fadi Ahwal, Simon B. Duckett, Richard O. John, Peter M. Richardson, Robin Virgo, Meghan E. Halse

**Affiliations:** 1Department of Chemistry, University of York, Heslington, York YO10 5DD, UK; fraser.hill-casey@york.ac.uk (F.H.-C.); aminata.sakho@york.ac.uk (A.S.); aaam508@york.ac.uk (A.M.); mr1303@york.ac.uk (M.R.); prichardson@ucsb.edu (P.M.R.); rv536@york.ac.uk (R.V.); 2Centre for Hyperpolarisation in Magnetic Resonance, University of York, Heslington, York YO10 5NY, UK; fadi.ahwal@york.ac.uk (F.A.); simon.duckett@york.ac.uk (S.B.D.); richard.john@york.ac.uk (R.O.J.)

**Keywords:** NMR spectroscopy, hyperpolarization, parahydrogen, zero-to-ultra-low-field (ZULF) NMR, signal amplification by reversible exchange (SABRE)

## Abstract

Hyperpolarization methods, which increase the sensitivity of nuclear magnetic resonance (NMR) and magnetic resonance imaging (MRI), have the potential to expand the range of applications of these powerful analytical techniques and to enable the use of smaller and cheaper devices. The signal amplification by reversible exchange (SABRE) method is of particular interest because it is relatively low-cost, straight-forward to implement, produces high-levels of renewable signal enhancement, and can be interfaced with low-cost and portable NMR detectors. In this work, we demonstrate an in situ approach to SABRE hyperpolarization that can be achieved using a simple, commercially-available Earth’s field NMR detector to provide ^1^H polarization levels of up to 3.3%. This corresponds to a signal enhancement over the Earth’s magnetic field by a factor of ε > 2 × 10^8^. The key benefit of our approach is that it can be used to directly probe the polarization transfer process at the heart of the SABRE technique. In particular, we demonstrate the use of in situ hyperpolarization to observe the activation of the SABRE catalyst, the build-up of signal in the polarization transfer field (PTF), the dependence of the hyperpolarization level on the strength of the PTF, and the rate of decay of the hyperpolarization in the ultra-low-field regime.

## 1. Introduction

Magnetic resonance is a powerful analytical technique with a wide range of applications from the use of nuclear magnetic resonance (NMR) spectroscopy for reaction monitoring in solutions to the use of magnetic resonance imaging (MRI) for clinical diagnosis. However, when compared to other analytical methods, such as optical spectroscopies and mass spectrometry, magnetic resonance suffers from low sensitivity. In a standard NMR or MRI experiment, only a small fraction (typically a few ppm) of the nuclei in the sample are observed. This fraction of observed nuclei is called the polarization. Polarization is directly proportional to the applied magnetic field strength, so the inherent insensitivity of NMR can be partially overcome through the use of strong and homogeneous magnetic fields. However, strong magnetic fields require expensive instrumentation that is non-portable and needs expert maintenance. An alternative approach to overcoming the sensitivity issue of NMR is hyperpolarization. Hyperpolarization is a general name for a range of techniques that aim to increase the sensitivity of magnetic resonance by temporarily increasing the fraction of observable nuclei [1,2,3,4,5,6,7]. We focus here on a hyperpolarization method that uses the singlet nuclear spin isomer of dihydrogen, called parahydrogen (p-H_2_), as the source of the NMR signal enhancement [8,9].

The idea of using p-H_2_ to generate hyperpolarization in NMR was first introduced by Bowers and Weitekamp in 1986 [10], and it is generally referred to by the term “parahydrogen-induced polarization (PHIP)” [11]. The key benefits of PHIP are that high levels of polarization (up to tens of percent) can be achieved, polarization levels are independent of the magnetic field strength used to detect the enhanced NMR response, and the instrumentation requirements are compact and low-cost when compared to other hyperpolarization methods. Parahydrogen can be used as a source of hyperpolarization because the two ^1^H nuclei (protons) in p-H_2_ exist in a pure single nuclear singlet state. This singlet state has no net angular momentum and so it is NMR-silent. However, if the symmetry of the pair of protons in p-H_2_ is broken by a chemical reaction, following pair-wise addition to a substrate or oxidative addition at a metal center, for example, the NMR signals for the p-H_2_-derived protons in the product molecule are enhanced, often by many orders of magnitude. In the original PHIP methods, called PASASDENA [12] and ALTADENA [13], the hyperpolarization is observed following a parahydrogenation reaction. This approach has been widely used for mechanistic studies in inorganic chemistry [14,15] and has been explored as a route to generating hyperpolarized contrast agents for MRI [1,16,17,18]. Hydrogenative PHIP has also been used to observe ^1^H, ^13^C, and ^15^N hyperpolarization in low and ultra-low magnetic fields, including in the Earth’s magnetic field and below [19,20,21,22,23].

A key limitation of the hydrogenative PHIP approach is that a suitable starting material is required and the reaction is irreversible. Therefore, it is inherently a single-shot approach. These limitations were overcome in 2009 by the introduction of the signal amplification by reversible exchange (SABRE) method by Duckett and co-workers [24,25]. SABRE is a non-hydrogenative version of PHIP that uses a transition metal complex to catalytically transfer the polarization from p-H_2_ to a target substrate without altering the chemical identity of the substrate. This transfer of polarization is mediated by the scalar (*J*) coupling network of the active SABRE catalyst and is most efficient in very weak magnetic fields in the range of 0–10 mT. A typical SABRE hyperpolarization experiment is carried out in two steps. First, the reversible exchange reaction with p-H_2_ is carried out over a few seconds in a weak magnetic field (nT – mT) in order to build-up hyperpolarization on the target substrate in solution. Second, the sample is rapidly transferred into a high-field NMR spectrometer (≥ 1 T) for signal detection. The practical implementation of this two-step process, either using the manual shaking of the sample in an NMR tube or using an automated flow approach [26,27,28,29], is straight-forward, but it limits our ability to directly observe the polarization transfer that is at the heart of the SABRE process. We note that in situ SABRE at a high field has been demonstrated using radio-frequency (RF) driven transfer [30]; however, we focus here on the spontaneous transfer of polarization that occurs in the low-field regime.

In this work, we use an in situ SABRE approach where the polarization transfer and NMR signal detection are achieved without the need to transport the sample between the low and high-field regimes. To achieve this, we use a switchable electromagnet to generate the required polarization transfer field (PTF) of a few mT, and then we detect the enhanced NMR response in the Earth’s magnetic field. We note that a similar approach was used by Hövner and co-workers to demonstrate continuous hyperpolarization and MRI using SABRE with Earth’s field detection [27,31]. In addition, SABRE hyperpolarization has been observed in low and ultra-low fields using standard detectors [32], superconducting quantum interference devices (SQUIDs) [33,34,35], and atomic magnetometers [36]. PHIP hyperpolarization has also been observed using in situ detection, where the polarization transfer and detection are both achieved in the mT regime [37,38]. Herein, we demonstrate how in situ SABRE using a simple, commercially available Earth’s field NMR spectrometer for detection, enables the direct interrogation of several important aspects of the SABRE experiment including the evolution of the SABRE signal enhancement during the catalyst activation, the build-up of hyperpolarization in the presence of the polarization transfer field, and the subsequent relaxation of the hyperpolarization in the ultra-low-field regime.

## 2. Results and Discussion

### 2.1. In Situ SABRE Hyperpolarization with H-EFNMR Detection

The SABRE process is illustrated schematically in Figure 1a. The SABRE catalyst is an octahedral iridium di-hydride complex that contains two substrate molecules (pyridine in this case) bound trans to the hydrides and a third substrate molecule bound trans to a stabilizing *N*-heterocyclic carbene (IMes = 1,3-bis(2,4,6-trimethyl-phenyl)-imidazolium) [39]. The hydrides and substrate molecules bound trans to the hydrides are in rapid reversible exchange with an excess of parahydrogen and substrate in free solution. The polarization transfer between the p-H_2_-derived hydrides and the NMR-active nuclei of the bound substrates is mediated by the scalar (*J*) coupling network of the iridium di-hydride complex. For optimal transfer to the ^1^H nuclei on the substrate, this exchange reaction is carried out in a weak magnetic field of around 6.5 mT [40,41,42]. A schematic of the experimental set-up for our in situ SABRE approach is shown in Figure 1b, where the exchange reaction is shown to have been carried out in solution within the reaction chamber. This reaction chamber sits within the probe of a commercial Earth’s field (EF) NMR spectrometer (Terranova-MRI, Magritek). On the outside of the EFNMR probe is an electromagnet that provided a polarization transfer field of 3.13 mT A^−1^. The probe also contains three orthogonal linear magnetic field gradients that are used for shimming to improve the local homogeneity of the Earth’s magnetic field, as well as a *B*_1_ coil for signal excitation and detection. The ^1^H-NMR resonance frequency in the Earth’s magnetic field for our experiments was approximately 2 kHz (*B_E_* ~ 50 µT). To initiate the SABRE exchange reaction, p-H_2_ is bubbled through the solution via a porous frit. The p-H_2_ flow rate is controlled by maintaining a pressure drop across the inlet and exhaust of the reaction chamber. In our experiments, the average pressure in the cell during bubbling was approximately 4 bar absolute. The inlet and exhaust pressures are defined by the user and controlled by a polarizer box, developed by Duckett and co-workers in collaboration with Bruker [26,43]. The source of parahydrogen was a commercial Bruker parahydrogen generator operating at a conversion temperature of 38 K.

Figure 2a presents the general pulse sequence for in situ SABRE hyperpolarization with ^1^H-EFNMR detection. In the first step, p-H_2_ bubbling is initiated and following a short delay, *d*_1_, the polarization transfer field (PTF) is switched on by passing a fixed current through the outer coil of the EFNMR probe. The duration of the PTF pulse, $\tau_{\mathrm{PTF}}$, is controlled by the user and limited by the resistive heating of the coil. Following the build-up of hyperpolarization along the axis of the EFNMR probe, the PTF is adiabatically switched-off to allow for the enhanced magnetization to re-orient along the Earth’s magnetic field. The minimum delay for the switching of the PTF is *d*_2_ = 100 ms. To achieve signal detection in the Earth’s magnetic field, an RF pulse, θ, is applied, and following an acquisition delay, typically *d*_3_ = 25 ms to allow for coil ring-down, the NMR signal is recorded. To complete the experiment, an optional delay, *d*_4_, is followed by the release of the p-H_2_ pressure. A delay of *d*_5_ = 3 s is included to allow time for the switch-off of the p-H_2_ bubbling and subsequent out-gassing of H_2_ from the solution. We note that this is essentially the same pulse sequence used to acquire non-SABRE-enhanced EFNMR spectra except that for a standard spectrum there is no bubbling of p-H_2_, and the polarization transfer field is used to pre-polarize the sample and is typically set to a field of 18.8 mT and applied for 4 s (see Section 2.2 for more details).

A representative single scan SABRE-enhanced ^1^H-EFNMR spectrum of 255 mM (82 µL) of pyridine with 5.1 mM (13 mg) of the SABRE catalyst in 4 mL of methanol is presented in Figure 2b, where PTF = 6.4 mT and $\tau_{\mathrm{PTF}}$ = 20 s. A 16 scan ^1^H-EFNMR spectrum of 562 mL of water is presented for comparison. The water spectrum is the average of 16 scans. These spectra clearly illustrate the significant benefits, in terms of both sensitivity and resolution, afforded by the in situ SABRE approach. The narrow linewidth of the SABRE-enhanced spectrum is primarily due to the smaller size of the SABRE sample (4 mL) compared to the water sample (562 mL), which leads to improved field homogeneity. Note, in these experiments, the entire sample was within the detection region of the coil. In addition, due to the temporal instability of the Earth’s magnetic field, signal averaging, which was used to increase the signal-to-noise ratio of the reference spectrum of water, can lead to peak broadening due to frequency shifts between successive scans. In general, there is no chemical shift information available in the Earth’s magnetic field (1 ppm = 0.002 Hz). To confirm that the enhanced ^1^H-EFNMR response was due exclusively to the hyperpolarized substrate, pyridine, we repeated the experiment using pyridine-d_5_ as the substrate. No ^1^H-EFNMR signal was observed. This confirms that for this system, the observed SABRE hyperpolarization is derived from the substrate, pyridine.

### 2.2. Calibration of SABRE Hyperpolarization with H-EFNMR Detection

In a standard SABRE experiment with high-field NMR detection, the level of signal enhancement is quantified as the ratio of the NMR signal measured with and without SABRE enhancement for the same sample. In our case, in the absence of SABRE hyperpolarization (i.e., without p-H_2_ bubbling) no ^1^H-EFNMR signal was observed for the SABRE sample containing 82 µL (255 mM) of pyridine and 5.1 mM of catalyst in 4 mL of methanol. Therefore, it was not possible to directly quantify the level of SABRE polarization that was achieved in these experiments, and, as such, a calibration sample had to be used.

The SABRE polarization, $P_{SABRE}$, can be determined using Equation (1), where $S_{SABRE}$ is the signal per mol of ^1^H for the SABRE spectrum, $S_{ref}$ is the signal per mol of ^1^H for the reference, $P_{ref}$ is the polarization level of the reference, and *C_ref_* is the calibration constant that relates the SABRE signal to the polarization level.
(1)$$P_{SABRE}=\frac{S_{SABRE}}{S_{ref}}P_{ref}=C_{ref}S_{SABRE},$$

In order to determine *C_ref_*, we must first measure $S_{ref}$ for a reference sample with a known polarisation level. To achieve this, we measured the ^1^H-EFNMR signal for a range of different volumes of water. In principle, the observed ^1^H-EFNMR signal should increase linearly with the number of ^1^H nuclei in the sample and hence the volume of the sample. However, as can be observed in Figure 3, while there is a linear increase in signal for small volumes, this trend did not hold as the volume increased. This was due primarily to the fact that we used a method called pre-polarization to boost the ^1^H-EFNMR signal for the water measurements.

The polarization of ^1^H nuclei at thermal equilibrium in a magnetic field *B_p_* is given by Equation (2), where γ is the gyromagnetic ratio of ^1^H, ℏ is the reduced Planck’s constant and *T* is temperature.
(2)$$P=\frac{\gamma B_{p}\hbar }{2k_{B}T},$$

The idea of pre-polarization is to build-up polarization in a field, *B_p_*, that is much stronger than the Earth’s magnetic field, *B_E_* ~ 50 µT. Due to the linear relationship between the NMR signal and polarization, this results in an increase in the EFNMR signal by a factor of the ratio of the polarization and detection fields. In principle, for $B_{p}$ = 18.8 mT, this provides a signal enhancement of $B_{p}/B_{E}\sim \;380$. In practice, however, a smaller signal gain was observed for larger volume samples. This was due to the inhomogeneity of the magnetic field generated by the pre-polarization coil. While using a larger sample provides more nuclei to be observed, these nuclei will experience a smaller pre-polarization effect. Therefore, in order to determine an accurate value of $S_{ref}$, only the initial linear region (inset in Figure 3) was included in the fit to determine $S_{ref}$ = 1.64 ± 0.17 µV mol^−1^. Another effect of the finite sample volume was that the larger samples experienced a more inhomogeneous Earth’s magnetic field, resulting in a decrease in the effective transverse relaxation time, *T*_2_*, and an increase in the spectral line-with. The reduction in *T*_2_* led to an increased loss of signal during the acquisition delay (*d*_3_ = 25 ms) for the larger sample volumes. For the measurements in Figure 3, the *T*_2_* was estimated for each spectrum, and a correction was applied to the signal to account for the signal decay during the acquisition delay.

In principle, the polarization value for the water calibration samples, $P_{ref}$, can be calculated from Equation (2) with *B_p_* = 18.8 mT. However, two additional factors must be taken into account to improve the accuracy of this calibration method. First, the build-up of the polarization in the pre-polarization field, *B_p_*, is driven by longitudinal (*T*_1_) relaxation. Therefore the polarization level as a function of the duration of the polarization pulse ($\tau_{PTF}$ in Figure 2a) is given by Equation (3), where *T*_1_,*_Bp_* is the longitudinal relaxation time of water in the pre-polarization field, *B_p_*.
(3)$$P_{ref}=\frac{\gamma B_{p}\hbar }{2k_{B}T}\left(1-\exp\left[-\frac{\tau_{PTF}}{T_{1,B_{p}}}\right]\right),$$

The second factor that needs to be taken into account is the decay of the polarization during the delay, *d*_2_, between the switching of the pre-polarization field and the application of the RF pulse (see Figure 2a). The polarization decay during this delay is characterized by the relaxation time, $T_{1,B_{E}}$. Therefore, the level of polarization observed in the reference measurements on water is given by Equation (4).
(4)$$P_{ref}=\frac{\gamma B_{p}\hbar }{2k_{B}T}\left(1-\exp\left[-\frac{\tau_{PTF}}{T_{1,B_{p}}}\right]\right)\exp\left[-\frac{d_{2}}{T_{1,B_{E}}}\right],$$

The relaxation times of water were measured to be $T_{1,B_{p}}=2.1\pm 0.2\;s$ and $T_{1,B_{E}}=\;2.3\pm 0.1\;s$. The calibration measurements were carried out at room temperature (*T* = 295 K) and with *d*_2_ = 100 ms and $\tau_{PTF}$ = 4 s. Accordingly, the reference polarization level was calculated using Equation (4) to be $P_{ref}=\;$(5.3 ± 0.5) × 10^−8^, and the calibration factor was calculated from Equation (1) to be $C_{ref}=\;$(3.2 ± 0.3) × 10^−8^ mol µV^−1^. Using this calibration factor, the average polarization level of pyridine in the spectrum in Figure 2b was estimated to be *P_SABRE_* = (0.27 ± 0.03)%. We note that the uncertainty value for the polarization given here represented a cumulative error of 11.1% due to the uncertainties associated with the measurements of $S_{ref}$, $T_{1,B_{p}}$, $T_{1,B_{E}}$, and *S_SABRE_*. However, there were many other potential sources of uncertainty in this calibration method, e.g., daily variation in the response of the Earth’s field NMR spectrometer due to field fluctuations, varying noise levels and differences in field homogeneity. Therefore, this level uncertainty in the polarization value is likely to be an underestimate.

In previous work, we measured the SABRE polarization transfer efficiency for a sample containing 26 mM of pyridine and 5.2 mM of the IrCl(COD)(IMes) pre-catalyst in methanol-*d*_4,_ using a manual shaking approach and with high-field (9.4 T, 400 MHz) detection, to be *E* = 6.5%, *E* = 6.0% and *E* = 3.9% for the ortho, meta and para proton resonances of pyridine, respectively [44]. This corresponded to an average efficiency per proton of *E* = 5.8%, where the efficiency is defined as the substrate polarization level that would be achieved using 100% p-H_2_ enrichment. Using our in situ SABRE approach with H-EFNMR detection, we measured a maximum polarization level per proton of *P_SABRE_* = (3.3 ± 0.4)% for a sample containing 13 mg (5.1 mM) of pre-catalyst and 8.2 µL (26 mM, 5eq relative to the catalyst) of pyridine in 4 mL of methanol-*d*_4_. We estimated our actual p-H_2_ enrichment level to be $P_{pH_{2}}\;\sim \;82\%.$ Following the method from reference [44], this corresponds to a SABRE polarization transfer efficiency of *E* = (4.3 ± 0.5)%. This represents a drop in efficiency of ~25% relative to the standard high-field approach. In previous work, it was found that the polarization transfer efficiency using a flow system, where the p-H_2_ was bubbled through the solution inside a reaction cell similar to the one used here, resulted in a drop in efficiency by a factor of 5–6 [45]. This was attributed to a combination of factors including the relative inefficiency of p-H_2_ mixing when using the bubbling approach compared to manual shaking in a 5 mm diameter NMR tube and the time taken to stop bubbling and flow the sample into the NMR spectrometer for detection. The results here show that by removing the sample transfer step, a similar SABRE efficiency could be achieved with the bubbling approach when compared to manual shaking and high-field detection. Furthermore, for samples with faster relaxation times, the in situ approach provides enhanced benefits by minimizing the loss of polarization between the SABRE build-up in the PTF and NMR detection. In the future, we expect that improving the p-H_2_ mixing within the in situ system will lead to further increases in the SABRE efficiency.

### 2.3. Catalyst Activation and SABRE Signal Reproducibility

The active SABRE catalyst (Figure 1a) is air sensitive and specific to the target substrate. Therefore, in a standard SABRE experiment, the sample is prepared using an air-stable pre-catalyst of the form Ir(NHC)(COD)(Cl), where COD = 1,5-cyclooctadiene, and the *N*-heterocyclic carbene (NHC) used herein was IMes = 1,3-*bis*(2,4,6-trimethyl-phenyl)-imidazolium [39]. This pre-catalyst was transformed into the active catalyst in the presence of an excess of the substrate and p-H_2_. The mechanism of this activation process for the pre-catalyst Ir(NHC)(COD)(Cl) is well described in the literature [46,47]. In the in situ SABRE method described above, a sample containing the pre-catalyst and an excess of the target substrate is injected into the reaction chamber (see Figure 1b). The transformation of the pre-catalyst into the active SABRE species is carried out in situ by bubbling p-H_2_ through the solution. Due to the weak nature of the Earth’s magnetic field, ^1^H-EFNMR spectra do not provide chemical shift resolution and no signal is observed from non-hyperpolarized species. Therefore, we could not directly explore the mechanism of the activation. However, our in situ approach does provide the opportunity to directly probe the build-up of observable SABRE hyperpolarization as a function of p-H_2_-bubbling time and hence catalyst activation.

Figure 4 shows the SABRE-enhanced ^1^H-EFNMR signal amplitude for a 255 mM solution of pyridine with 5.1 mM of the SABRE pre-catalyst in 4 mL of methanol as a function of p-H_2_ bubbling time. Each point corresponds to a single repetition of the pulse sequence in Figure 2a with PTF = 6.4 mT and $\tau_{\mathrm{PTF}}$ = 20 s. The bubbling of p-H_2_ was stopped and re-started between each experiment to allow for the out-gassing of the normal H_2_ from solution and the dissolution of fresh p-H_2_. As expected, the SABRE-enhanced ^1^H-EFNMR signal showed an initial increase, which we attributed to the formation of the active catalytic species and a corresponding increase in the efficiency of the SABRE hyperpolarization transfer. After approximately 20 min of p-H_2_ bubbling, the enhanced signal reached a plateau value, indicating that the activation process had gone to completion. An average of the subsequent 105 repeat experiments yielded an average signal amplitude of 418 ± 7 µV. This corresponds to an average polarization level of *P_SABRE_* = (0.27 ± 0.03)%. The standard deviation across the repeat measurements was 1.6%, indicating a high level of reproducibility for this in situ SABRE approach. We note that for bubbling times greater than 1 h, a slow decrease in SABRE signal intensity was observed. This was associated with a gradual loss of solvent due to evaporation. It is well established in the literature that the efficiency of SABRE decreases as the concentration of substrate relative to the catalyst and p-H_2_ increases [47]. It has been suggested that this is due to the mechanism of the reversible exchange of the substrate and the parahydrogen on the SABRE catalyst. Due to the competition between these exchange processes, if the concentration of substrate is increased relative to the concentration of p-H_2_, the rate of parahydrogen exchange decreases, leading to a drop in the efficiency of the SABRE transfer process [26,47,48].

### 2.4. SABRE Polarization Build-Up and Decay

The in situ SABRE approach allowed for the observation of the build-up of the hyperpolarization in the PTF and the subsequent relaxation decay in the ultra-low-field regime. Figure 5a presents the build-up of ^1^H SABRE hyperpolarization for 255 mM pyridine (with 5.1 mM catalyst) in PTF = 6.4 mT as a function of the PTF duration. Figure 5b shows the subsequent decay of the hyperpolarization as a function of the delay (*d*_2_ in Figure 2a) between the PTF pulse and NMR detection. In the previous examples, protonated methanol (CH_3_OH) was used as the solvent. Due to the extremely low sensitivity of EFNMR detection, no NMR signal was observed from the protons in the solvent in the absence of SABRE hyperpolarization (i.e., in the absence of p-H_2_ bubbling). Therefore, unlike for high-field NMR, there was no need to use deuterated solvents. Nevertheless, it is interesting to investigate the effect of a protonated versus deuterated solvent on the SABRE process. Accordingly, the black circles in Figure 5 represent SABRE experiments carried out in protonated methanol, and the gray circles represent SABRE experiments carried out in deuterated methanol.

The relaxation curves show a clear difference between the two solvents. The relaxation time for pyridine in deuterated methanol was *T*_1_ = 10.3 ± 0.3 s, while the relaxation time in protio methanol was *T*_1_ = 6.00 ± 0.04 s. This reduction in relaxation time was attributed to the increased dipole–dipole interactions between the ^1^H nuclei on the substrate and in the solvent. These results indicate that in the Earth’s magnetic field, any ^2^H quadrupolar relaxation effects are much weaker than the effects of the ^1^H–^1^H dipole–dipole relaxation. This is in agreement with previous work by Duckett and co-workers, which showed that partial deuteration of both the substrate and the SABRE catalyst increased the efficiency of SABRE polarization transfer in a PTF of 6.5 mT [48]. We note that other groups have found that, in the sub-Earth’s field regime, the presence of quadrupolar nuclei, such as ^14^N, leads to a reduction in the efficiency of hyperpolarization transfer to ^13^C, an effect that is attributed to strong quadrupolar relaxation effects in this field regime [49].

While there are clear differences in the hyperpolarization lifetimes of pyridine in the two solvents, the results in Figure 5a suggest that the effect of deuteration of the solvent on the build-up time of the hyperpolarization was not significant in this case. A similar constant polarization level was reached after approximately 20 s in both solvents.

The final aspect of the SABRE process that we explored using the in situ SABRE approach was the effect of the PTF on the efficiency of the SABRE polarization transfer. Figure 6 presents the polarization level as a function of PTF for three substrates: pyridine, pyrazine and isonicotinamide. In all three cases, the form of the PTF curve was very similar, with a maximum of around 6.5 mT. This is in agreement with high-field SABRE measurements for similar substrates [50] and with the accepted theoretical picture of SABRE polarization transfer, whereby polarization transfer efficiency is maximized when the difference in frequency (in Hz) between the hydrides and the ^1^H nuclei of the bound substrate is approximately equal to the dominant scalar (*J*) coupling in the network [40,41,42]. This is typically the ^1^H–^1^H coupling between the hydrides, which is on the order of 8–10 Hz. For a difference in chemical shift between the hydrides (~ −23 ppm) and the ^1^H nuclei in the substrate (~7 ppm), of 30 ppm a field of 6.5 mT fulfils this resonance condition and maximizes polarization transfer.

## 3. Materials and Methods

### 3.1. Sample Preparation

Samples for the hyperpolarization of pyridine were prepared by dissolving 13 mg (5.1 mM) of the SABRE pre-catalyst [IrCl(COD)(IMes)] (where COD = 1,5 cyclooctadiene and IMes = 1,3-bis(2,4,6-trimethylphenyl)-imidazol-2-ylidine) in 4 mL of solvent (methanol or methanol-d_4_, as indicated in the text). The sample was sonicated for a few minutes until homogeneous. Pyridine was then added to the solution (82 µL (255 mM, 50 eq relative to the pre-catalyst) or 8.2 µL (26 mM, 5 eq relative to the pre-catalyst), as indicated in the text). For the PTF curves presented in Figure 6, a larger sample volume of 5 mL was used. Accordingly, these samples were prepared using 16 mg (5.0 mM) of the SABRE pre-catalyst and 250 mM (50 equivalents relative to the pre-catalyst) of the substrate. The substrates: pyridine, pyrazine, and nicotinamide (Scheme 1), were purchased from Sigma-Aldrich (St. Louis, MO, USA) and used without further modification. The pre-catalyst was synthesized in-house. At the start of each experimental session, the solution containing the pre-catalyst and the substrate was injected into the reaction chamber through the p-H_2_ inlet using a syringe. The total volume of the solution injected was either 4 or 5 mL, as indicated in the text.

### 3.2. Parahydrogen Generation and Control

The source of hydrogen was a Precision Hydrogen Trace 500 electrolytic hydrogen generator (Peak Scientific, Inchinnan, UK) that produces hydrogen gas at 6 bar above atmospheric pressure. The conversion of H_2_ into p-H_2_ was achieved using a commercial parahydrogen generator (Bruker, Billerica, MA, USA) that passes the H_2_ through a toroidal path over a paramagnetic catalyst at a conversion temperature of 38 K. The p-H_2_ enrichment was estimated using high-field (500 MHz) ^1^H-NMR to be 82%. The flow of p-H_2_ through the reaction chamber was controlled by a commercial pneumatic control unit (Bruker, Germany), as shown in Figure 1b. The inlet and outlet pressure during p-H_2_ bubbling were maintained at 3.2 and 2.8 bar above atmospheric pressure, respectively. Between each SABRE experiment, the pressure in the reaction chamber was reduced to atmospheric pressure, allowing for the out-gassing of H_2_ from solution.

### 3.3. SABRE Hyperpolarization and EFNMR Detection

Earth’s field NMR detection was carried out using a Terranova MRI Earth’s field (EF) spectrometer (Magritek, Aachen, Germany). The local Earth’s magnetic field for these experiments was approximately *B_E_* = 42 µT, with a corresponding ^1^H Larmor frequency of 1788 Hz. The EFNMR apparatus was composed of a probe containing an RF (*B*_1_) coil, three orthogonal magnetic field gradient coils, an offset field coil, and a pre-polarizing field coil. Due to the very low ^1^H Larmor frequency, the EFNMR spectra are sensitive to a wide range of external sources of noise. One significant contribution came from the harmonics produced via the mains electricity, which, in the United Kingdom, generates noise peaks at odd integer multiples of 50 Hz. To avoid interference between the NMR signal and these periodic noise peaks, the offset coil was used to provide an additional homogeneous static magnetic field of 5.4 µT to shift the ^1^H Larmor frequency to approximately 2020 Hz. This offset field was applied throughout the entire pulse sequence, along with the linear magnetic field gradients that were used for first-order shimming. The EFNMR spectrometer and associated NMR experiments were controlled via a PC running the Terranova-Expert software package within Prospa (Magritek, Germany). The cylindrical reaction chamber (i.d. = 24 mm; o.d. = 28 mm; *L* = 24 mm) was manufactured in-house and was comprised of a Simax glass cell (Kavalier, Prague, Czech Republic). The cell includes an inlet port and an outlet port of o.d = 4 mm. Connections between the reaction cell and the gas manifold system were made via 1/16” PEEK fittings (IDEX, Lake Forest, IL, USA). The p-H_2_ was bubbled through a VitraPOR frit (Robu, Hattert, Germany) with an approximate pore size distribution of 40–100 µm.

The pulse sequence for all EFNMR experiments is presented in Figure 2a. Unless otherwise stated in the text, the following delay parameters were used: *d*_1_ = *d*_4_ = 0 s, *d*_2_ = 100 ms, *d*_3_ = 25 ms, *d*_5_ = 3 s, and a 90° RF excitation pulse with an amplitude of 0.3 V and a duration of 28.9 ms was applied for signal excitation. For the non-SABRE-enhanced ^1^H-EFNMR spectra of water, a pre-polarization field of 18.8 mT, produced by passing a current of 6 A through the electromagnet on the outside of the EFNMR probe, was applied for 4 s. The relaxation times of the water samples were measured to be $T_{1,B_{p}}$= 2.2 ± 0.2 s and $T_{1,B_{E}}$= 2.3 ± 0.1 s using pseudo-2D experiments where the ^1^H-EFNMR signal amplitude was measured as a function of varying either the duration of the pre-polarizing pulse, $\tau_{PTF},$ ($T_{1,B_{p}}$) or the delay between the polarizing pulse and the RF pulse, *d*_2_ ($T_{1,B_{E}}$). For the SABRE-enhanced EFNMR experiments, the p-H_2_ bubbling was initiated by passing a command to the polarizer (see Figure 1b) from the controlling PC using RS232 serial communication within the Terrnaova-Expert software. Immediately following the sending of the signal to start bubbling, the pulse sequence was initiated using the digital signal processor (DSP) of the Terranova spectrometer. Maximum signals were observed for *d*_1_ = 0 s, indicating that the finite time taken (tens of ms) between the serial communication and the initialization of the DSP was sufficient to allow for bubbling to commence before turning on the PTF. Unless otherwise specified, a PTF = 6.4 mT (2.05 A) was applied for a duration $\tau_{PTF}$ = 20 s. Due to the narrow linewidths and, hence, long *T*_2_* values, of the SABRE ^1^H-EFNMR spectra, a long acquisition time between *t_acq_* = 5 s and *t_acq_* = 7 s was used. All ^1^H-EFNMR spectra were zero-filled by a factor of 4, Fourier transformed, phased, and integrated using a home-written macro within Prospa (Magritek, Germany). Consistent acquisition and processing parameters, including zero-filling, were used throughout to enable direct comparisons between signal integrals, measured in µV, from different experiments. To determine the *T*_2_* for the water reference measurements, a 80 Hz region centered at the Larmor frequency was extracted from each-EFNMR spectrum and inverse Fourier transformed. The resultant signal decay was fit to a single exponential decay using the curve fitting tool in MATLAB to determine the *T*_2_* value (see Table 1.). The following correction was then applied to each signal integral, where *d*_3_ = 25 ms is the acquisition delay, as defined in Figure 2a.
(5)$$S_{corr}=S_{0}/\exp\left(-d_{3}/T_{2}^{*}\right)$$

At the start of each experimental session, the EFNMR parameters including *x*, *y*, and *z* shims, the offset coil current and the tuning capacitance were optimized using the ^1^H-EFNMR signal from a 562 mL bottle of water. Subsequent SABRE experiments were initiated by injecting the sample into the reaction chamber located in the center of the EFNMR probe. The catalyst activation was achieved using the procedure discussed in Section 2.1 in the text (see Figure 4) and was considered complete once a consistent level of polarization had been reached (e.g., after 20 min in Figure 4). Between samples, the reaction chamber was cleaned by flushing once with methanol and once with acetone before drying under a flow of N_2_. The results presented in Figure 6 were obtained with an earlier version of the reaction chamber with a comparable design but a larger volume. These experiments were carried out using 5 mL samples, as noted in the text.

All of the NMR data presented in this work are freely available through the York Research Database.

## 4. Conclusions

In this work, we demonstrated an in situ SABRE hyperpolarization method that uses a simple, commercially available Earth’s field NMR spectrometer to detect the enhanced NMR response. We observed a maximum SABRE hyperpolarization level of 3.3% for a 4 mL sample containing 26 mM of pyridine and 5.1 mM of the SABRE catalyst in methanol-*d*_4_. This represents an NMR signal enhancement factor of ε = 2.3 × 10^8^ over thermal equilibrium in the Earth’s magnetic field and a factor of ε = 5.0 × 10^5^ over thermal equilibrium in the pre-polarization field of 18.8 mT. Measurements performed in CH_3_OH and CD_3_OD provided similar levels of the SABRE-enhanced ^1^H-EFNMR signal. In addition to the benefits of increased sensitivity, the use of SABRE was shown to improve the ^1^H-EFNMR linewidths due to the smaller magnetic field inhomogeneity experienced by the reduced sample volume. After the activation of the SABRE pre-catalyst, the SABRE-enhanced NMR response was found to be highly reproducible, with a standard deviation of 1.6% over 105 repeat measurements. Having demonstrated the high sensitivity and reproducibility of the in situ SABRE method, we applied this approach to explore the polarization transfer process in the low-field regime, including the build-up of polarization as a function of the amplitude and duration of the PTF and the quantification of the rate of decay of the hyperpolarization in the Earth’s magnetic field. These direct measurements allow for a comparison of these effects between different solvents, substrates, catalysts and reaction conditions without the confounding effects associated with transporting the sample between the PTF and the NMR detector. Our field-cycling approach with detection in the Earth’s magnetic field has the benefit of allowing for the direct comparison of experiments in different PTFs. However, it would also be of interest to extend this work to include direct detection in the PTF, as this would remove any effects from field switching. In addition, working at a higher RF frequency would improve the signal-to-noise by reducing sensitivity to external noise. This approach was not taken here because the PTF provided by the EFNMR spectrometer was not sufficiently homogeneous for NMR detection. However, designs for NMR spectrometers operating in the mT regime are available in the literature and have been used for in situ detection of PHIP [37,38]. In this work, we focused on N-heterocycles as substrates, a single SABRE pre-catalyst, IrCl(COD)(IMes), and methanol as the solvent. However, the high sensitivity and resolution achieved herein suggest that a wider range of SABRE systems will be amenable to study using this in situ approach. We expect that this will be a useful and practical tool to further optimize the SABRE process in the future by developing a better understanding of the behavior in the low-field regime.

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
