# Peer review of "In Situ SABRE Hyperpolarization with Earth’s Field NMR Detection"

_molecules, 2019, doi:10.3390/molecules24224126_

Round 1

Reviewer 1 Report

This paper looks to be scientifically correct, and to the best of my knowledge, nobody has done a 6.5 mT SABRE experiment, and then transferred the field (as opposed to the sample) down to Earth's field reasonably quickly.  

The outstanding question is, why would anyone want to do that?  There are interesting applications of Earth-field NMR, but almost none of them make sense with an exogenous polarized agent (at best, you can imagine using a pulsed field to prepolarize water).  The reported polarization levels are very low for SABRE (you don't very much beat water) but even if you did, other than for a teaching experiment, it is not clear what the advantages are.  It is true that you can monitor the dynamics by earth's field NMR, but this is fundamentally no different than doing it at any other non-resonant field strength (except perhaps for the hope that the field would be moderately homogeneous without shimming).

So I think the work is publishable, but the significance is modest.

Author Response

Reviewer #1

The outstanding question is, why would anyone want to do that?  There are interesting applications of Earth-field NMR, but almost none of them make sense with an exogenous polarized agent (at best, you can imagine using a pulsed field to prepolarize water).  The reported polarization levels are very low for SABRE (you don't very much beat water) but even if you did, other than for a teaching experiment, it is not clear what the advantages are.  It is true that you can monitor the dynamics by earth's field NMR, but this is fundamentally no different than doing it at any other non-resonant field strength (except perhaps for the hope that the field would be moderately homogeneous without shimming).

Response:

The goal of this work is not in the first instance to use SABRE to enable new applications of EFNMR itself but rather as a relatively inexpensive and straightforward method to probe the SABRE technique in the regime in which the transfer takes place without the need for the shuttling of the sample between the high and low-field regime. Therefore, we expect that the main impact of the work would be in improving our understanding of SABRE and further optimising this promising hyperpolarisation technique for a wide range of applications, which are by no means limited to EFNMR.

Reviewer 2 Report

This manuscript describes the in situ detection of SABRE hyperpolarization process by performing classical SABRE hyperpolarization in a few milli-Tesla range, followed by magnetic field ramp to the Earth’s magnetic field (ca. 50 micro-Tesla) followed by the detection in the Earth’s magnetic field at ca. 2 kHz resonance frequency. Overall, this is a nicely conducted work which certainly deserves a publication in this special issue. The work is conducted with care, and it will be of interest to a growing field of SABRE in particular and field of hyperpolarization in general. Provided below are relatively small comments and suggestions aimed to improve the manuscript quality and put it in the context of previously published works.

In situ detection is obviously of interest for SABRE hyperpolarization, because the classic 1H-to-1H spontaneous polarization is optimal in the field of a few milli-Tesla, which can be conveniently created by the electromagnet. The authors employ a commercially available equipment, which is a clear strength of the presented paper. Nevertheless, I find particularly awkward that the authors ramp the field to milli-Tesla range for SABRE to happen, and instead of detection at that high-field, they ramp the field down to micro-Tesla range. I completely understand the hardware limitations of their non-homogeneous pre-polarization magnet coil; however, the electromagnet designs and corresponding RF probes in a few milli-Tesla range have certainly been described: (e.g. Coffey, A. M.; Shchepin, R. V.; Truong, M. L.; Wilkens, K.; Pham, W.; Chekmenev, E. Y., Open-Source Automated Parahydrogen Hyperpolarizer for Molecular Imaging Using 13C Metabolic Contrast Agents. Chem. 2016, 88 (16), 8279-8288. And Coffey, A. M.; Shchepin, R. V.; Feng, B.; Colon, R. D.; Wilkens, K.; Waddell, K. W.; Chekmenev, E. Y., A Pulse Programmable Parahydrogen Polarizer Using a Tunable Electromagnet and Dual Channel NMR Spectrometer. J. Magn. Reson. 2017, 284, 115-124), and they have been employed for in-situ detection of PHIP in milli-Tesla magnetic field range. The key advantage of this approach (demonstrated in Ref. #31) is the direct detection of SABRE signal with sub-second temporal resolution meaning that the build-up and decay kinetics can be recorded on a single pass. The limitations of the existing setup should be clearly described (which in part is already nicely done in the manuscript) in the context of the alternative approach. Moreover, EMF shielding at higher frequencies is significantly easier to accomplish (which in fact was a problem in the current study). Although the authors perform in-situ detection, it happens with a lag time and at a different field. I am not sure I fully agree with Equation 1. Pref comes from the pre-polarization at 18.8 mT, which would be a higher polarization value, which the authors described later in the text. That’s fine. However, besides the signal (per unit concentrations) and equilibrium proton polarization (at any field), once the RF pulse is applied, T2* results in the signal loss during delay d2 (25 ms). Because T2* values are different for both samples (Reference and HP sample), this delay will have disproportionately larger effect on water sample. While the correction will likely be relatively small (a few percent), it is still worth considering it. The corresponding math has been described elsewhere: Nikolaou, P.; Coffey, A. M.; Ranta, K.; Walkup, L. L.; Gust, B.; Barlow, M. J.; Rosen, M. S.; Goodson, B. M.; Chekmenev, E. Y., Multi-Dimensional Mapping of Spin-Exchange Optical Pumping in Clinical-Scale Batch-Mode 129Xe Hyperpolarizers. Phys. Chem. B 2014, 118 (18), 4809–4816. In ideal world, the authors should have compared the polarization in their sample with a different detection method: e.g. higher-field spectrometer (e.g. Nikolaou, P.; Coffey, A. M.; Walkup, L. L.; Gust, B. M.; Whiting, N.; Newton, H.; Barcus, S.; Muradyan, I.; Dabaghyan, M.; Moroz, G. D.; Rosen, M.; Patz, S.; Barlow, M. J.; Chekmenev, E. Y.; Goodson, B. M., Near-unity nuclear polarization with an 'open-source' 129Xe hyperpolarizer for NMR and MRI. Natl. Acad. Sci. U. S. A. 2013, 110 (35), 14150-14155.) That would ensure that whatever polarization values they are measuring actually agree with what is claimed. I understand that this is challenging to do, b/c the same needs to be transferred, but this could be done in principle. The non-initiated reader may wonder why this was not done, and the authors should provide a corresponding explanation. Figure 1 would certainly benefit from the schematic of the manifold employed in this study. At the moment only the overall scheme is provided – lacking details for those who would like to reproduce these results. The description of the reaction vessel (or part number ideally) should also be provided. The authors should also emphasize that most text pertains to spontaneous SABRE. In case if the RF-SABRE is employed, it already employs in situ detection so to speak (e.g. Pravdivtsev, A. N.; Yurkovskaya, A. V.; Vieth, H.-M.; Ivanov, K. L., Spin mixing at level anti-crossings in the rotating frame makes high-field SABRE feasible. Chem. Chem. Phys. 2014, 16, 24672-24675.) The exponential decay is nicely simulated in Figure 5. What were the exponential build-up constants in Figure 5a? How do they correlate with T1 values derived from Figure 5b?

Author Response

1. In situ detection is obviously of interest for SABRE hyperpolarization, because the classic 1H-to-1H spontaneous polarization is optimal in the field of a few milli-Tesla, which can be conveniently created by the electromagnet. The authors employ a commercially available equipment, which is a clear strength of the presented paper. Nevertheless, I find particularly awkward that the authors ramp the field to milli-Tesla range for SABRE to happen, and instead of detection at that high-field, they ramp the field down to micro-Tesla range. I completely understand the hardware limitations of their non-homogeneous pre-polarization magnet coil; however, the electromagnet designs and corresponding RF probes in a few milli-Tesla range have certainly been described: (e.g. Coffey, A. M.; Shchepin, R. V.; Truong, M. L.; Wilkens, K.; Pham, W.; Chekmenev, E. Y., Open-Source Automated Parahydrogen Hyperpolarizer for Molecular Imaging Using 13C Metabolic Contrast Agents. Chem. 2016, 88 (16), 8279-8288. And Coffey, A. M.; Shchepin, R. V.; Feng, B.; Colon, R. D.; Wilkens, K.; Waddell, K. W.; Chekmenev, E. Y., A Pulse Programmable Parahydrogen Polarizer Using a Tunable Electromagnet and Dual Channel NMR Spectrometer. J. Magn. Reson. 2017, 284, 115-124), and they have been employed for in-situ detection of PHIP in milli-Tesla magnetic field range. The key advantage of this approach (demonstrated in Ref. #31) is the direct detection of SABRE signal with sub-second temporal resolution meaning that the build-up and decay kinetics can be recorded on a single pass. The limitations of the existing setup should be clearly described (which in part is already nicely done in the manuscript) in the context of the alternative approach. Moreover, EMF shielding at higher frequencies is significantly easier to accomplish (which in fact was a problem in the current study).

Response

Our goal here was to use a simple commercially available device to probe the SABRE technique in situ and without the need to transport the sample between the polarisation build-up and signal detection stages. Direct detection in the PTF is not possible using this instrument, as the reviewer notes, due to the inhomogeneity of the PTF field. Designs for sufficiently homogeneous fields have been demonstrated for PHIP hyperpolarisation and could be used for SABRE. Such experiments would be complementary to the approach taken here. The benefit of the field cycling approach is that it allows for direct comparisons between experiences in different PTF fields with the same detection conditions. However, detection in the PTF itself would remove any effects from changing the field between polarisation build-up and detection and so provide additional insight. We have added a brief discussion on this point with the appropriate references to the conclusions.

2. Although the authors perform in-situ detection, it happens with a lag time and at a different field. I am not sure I fully agree with Equation 1. Pref comes from the pre-polarization at 18.8 mT, which would be a higher polarization value, which the authors described later in the text. That’s fine. However, besides the signal (per unit concentrations) and equilibrium proton polarization (at any field), once the RF pulse is applied, T2* results in the signal loss during delay d2 (25 ms). Because T2* values are different for both samples (Reference and HP sample), this delay will have disproportionately larger effect on water sample. While the correction will likely be relatively small (a few percent), it is still worth considering it. The corresponding math has been described elsewhere: Nikolaou, P.; Coffey, A. M.; Ranta, K.; Walkup, L. L.; Gust, B.; Barlow, M. J.; Rosen, M. S.; Goodson, B. M.; Chekmenev, E. Y., Multi-Dimensional Mapping of Spin-Exchange Optical Pumping in Clinical-Scale Batch-Mode 129Xe Hyperpolarizers. Phys. Chem. B 2014, 118 (18), 4809–4816. In ideal world, the authors should have compared the polarization in their sample with a different detection method: e.g. higher-field spectrometer (e.g. Nikolaou, P.; Coffey, A. M.; Walkup, L. L.; Gust, B. M.; Whiting, N.; Newton, H.; Barcus, S.; Muradyan, I.; Dabaghyan, M.; Moroz, G. D.; Rosen, M.; Patz, S.; Barlow, M. J.; Chekmenev, E. Y.; Goodson, B. M., Near-unity nuclear polarization with an 'open-source' 129Xe hyperpolarizer for NMR and MRI. Natl. Acad. Sci. U. S. A. 2013, 110 (35), 14150-14155.) That would ensure that whatever polarization values they are measuring actually agree with what is claimed. I understand that this is challenging to do, b/c the same needs to be transferred, but this could be done in principle. The non-initiated reader may wonder why this was not done, and the authors should provide a corresponding explanation.

Response:

Given the nature of the EFNMR measurements, calibration of an absolute polarisation level is extremely challenging. As acknowledged in the discussion of our calibration method, the uncertainty in the polarisation values produced is much higher than the error estimate based on our reference measurement uncertainties alone. This is due to the range of sources of variability that are challenging to capture in any reference calibration method of this type. However we agree with the reviewer that one source of error that we did not explicitly include is the signal loss due to T2* during the 25 ms dead-time of the B1 coil. To address this we have reanalysed the water calibration measurements and used measurements of the T2* for each sample to correct the signal values for T2* decay during the probe dead-time. The new calibration curve (new Figure 3) has a slope of 1.64 uV/mol giving a calibration constant  of (3.2 ± 0.3) x 10-8 mol µV-1. We note that accounting for T2* in this way has improved the non-linearity of the calibration curve for high volumes of water because the higher volumes have shorter T2* values leading to a larger correction. The polarisation numbers and figures have been updated accordingly and an explanation of this additional correction has been included in the text.

3. Figure 1 would certainly benefit from the schematic of the manifold employed in this study. At the moment only the overall scheme is provided – lacking details for those who would like to reproduce these results.

Response:

The manifold used to control the flow of pH2 is a unit sold by Bruker and therefore we are unable to provide a full schematic for this commercial device. References to the first use of this flow control unit in the literature are provided in the text. No modifications have been made for the purpose of this work.

4. The description of the reaction vessel (or part number ideally) should also be provided.

Response:

Additional details on the reaction vessel have been included in the experimental section.

5. The authors should also emphasize that most text pertains to spontaneous SABRE. In case if the RF-SABRE is employed, it already employs in situ detection so to speak (e.g. Pravdivtsev, A. N.; Yurkovskaya, A. V.; Vieth, H.-M.; Ivanov, K. L., Spin mixing at level anti-crossings in the rotating frame makes high-field SABRE feasible. Chem. Chem. Phys. 2014, 16, 24672-24675.)

Response:

A statement clarifying this point and the reference has been added to the introduction.

6. The exponential decay is nicely simulated in Figure 5. What were the exponential build-up constants in Figure 5a? How do they correlate with T1 values derived from Figure 5b?

Response:

The build-up curves do not follow a single exponential and so no fits have been included. The relationship between T1 decay and the build-up rate is not well established and is currently under investigation.

Reviewer 3 Report

This is a very well written manuscript about performing SABRE hyperpolarization in low field and acquiring the spectrum in Earth's magnetic field after switching off the polarization field.  This technique eliminates the need of transferring the sample from low field used for polarization to high field for signal acquisition.  I do not have any questions or concerns about this work.  Publication is recommended essentially in its present form.

Just a few notes:

Could the spectra be recorded at the same field that was used for polarization?  This could further simplify the design.

Is the SABRE process affected by magnetic field inhomogeneities?

Why is the polarization of isonicotinamide so much lower (one third) than that of pyridine and pyrazole (Figure 6), when the chemical structure of these substrates is very similar?

Author Response

1. Could the spectra be recorded at the same field that was used for polarization?  This could further simplify the design.

Response

A discussion of this point has been added (see response to Reviewer # 2)

2. Is the SABRE process affected by magnetic field inhomogeneities?

Response:

As can be observed in the PTF curves in Figure 6, the SABRE polarisation transfer field condition is quite broad and so the inhomogeneity across the relatively small reaction cell is not expect to significantly affect the efficiency of the SABRE transfer.

3. Why is the polarization of isonicotinamide so much lower (one third) than that of pyridine and pyrazole (Figure 6), when the chemical structure of these substrates is very similar?

Response:

The SABRE efficiency is a complex  product of a wide range of factors including the kinetics of the chemical exchange, relaxation times and the coupling network within the active catalyst and the substrate. One of the motivations of this work is to use the in situ approach to explore these factors to better understand and predict variations in efficiency such as this.